# Significance of the Modified NUTRIC Score for Predicting Clinical Outcomes in Patients with Severe Community-Acquired Pneumonia

**DOI:** 10.3390/nu14010198

**Published:** 2021-12-31

**Authors:** Chia-Cheng Tseng, Chih-Yen Tu, Chia-Hung Chen, Yao-Tung Wang, Wei-Chih Chen, Pin-Kuei Fu, Chin-Ming Chen, Chih-Cheng Lai, Li-Kuo Kuo, Shih-Chi Ku, Wen-Feng Fang

**Affiliations:** 1Division of Pulmonary and Critical Care Medicine, Department of Internal Medicine, Kaohsiung Chang Gung Memorial Hospital, Chang Gung University College of Medicine, Kaohsiung 83301, Taiwan; cctseng@hotmail.com.tw; 2Department of Respiratory Care, Chang Gung University of Science and Technology, Chiayi 61363, Taiwan; 3Division of Pulmonary and Critical Care Medicine, Department of Internal Medicine, China Medical University Hospital, Taichung 404, Taiwan; chesttu@gmail.com (C.-Y.T.); hsnu758@gmail.com (C.-H.C.); 4Division of Pulmonary Medicine, Department of Internal Medicine, Chung Shan Medical University Hospital, Taichung 402, Taiwan; cshy1025@gmail.com; 5School of Medicine, Chung Shan Medical University, Taichung 402, Taiwan; 6Department of Chest Medicine, Taipei Veterans General Hospital, Taipei 11217, Taiwan; wiji.chen@gmail.com; 7Department of Critical Care Medicine, Taichung Veterans General Hospital, Taichung 407219, Taiwan; yetquen@gmail.com; 8Program in Translational Medicine, National Chung Hsing University, Taichung 402010, Taiwan; 9Department of Intensive Care Medicine, Chi Mei Medical Centre, Tainan 710, Taiwan; chencm3383@gmail.com; 10Department of Internal Medicine, Kaohsiung Veterans General Hospital, Tainan Branch, Tainan 71051, Taiwan; dtmed141@gmail.com; 11Division of Pulmonary and Critical Care Medicine, MacKay Memorial Hospital, Taipei 10449, Taiwan; lmn4093@gmail.com; 12Division of Chest Medicine, Department of Internal Medicine, National Taiwan University Hospital, Taipei 100, Taiwan; scku1015@gmail.com

**Keywords:** nutrition, severe community-acquired pneumonia, clinical outcome

## Abstract

Nutritional status could affect clinical outcomes in critical patients. We aimed to determine the prognostic accuracy of the modified Nutrition Risk in Critically Ill (mNUTRIC) score for hospital mortality and treatment outcomes in patients with severe community-acquired pneumonia (SCAP) compared to other clinical prediction rules. We enrolled SCAP patients in a multi-center setting retrospectively. The mNUTRIC score and clinical prediction rules for pneumonia, as well as clinical factors, were calculated and recorded. Clinical outcomes, including mortality status and treatment outcome, were assessed after the patient was discharged. We used the receiver operating characteristic (ROC) curve method and multivariate logistic regression analysis to determine the prognostic accuracy of the mNUTRIC score for predicting clinical outcomes compared to clinical prediction rules, while 815 SCAP patients were enrolled. ROC curve analysis showed that the mNUTRIC score was the most effective at predicting each clinical outcome and had the highest area under the ROC curve value. The cut-off value for predicting clinical outcomes was 5.5. By multivariate logistic regression analysis, the mNUTRIC score was also an independent predictor of both clinical outcomes in SCAP patients. We concluded that the mNUTRIC score is a better prognostic factor for predicting clinical outcomes in SCAP patients compared to other clinical prediction rules.

## 1. Introduction

Severe community-acquired pneumonia (SCAP) remains a frequent cause of hospital admission and is the most common reason for sepsis needing intensive care unit (ICU) admission [1,2]. Although there have been recent advances in supportive care and early diagnosis of community-acquired pneumonia (CAP) can prompt effective antibiotic administration, SCAP still contributes to substantial morbidity and mortality, more frequently seen in elderly patients and those with considerable comorbidities [3]. Clinically, it is still important to identify a major rule for predicting clinical outcomes in patients with SCAP.

Many clinical prediction rule scoring systems are used to evaluate the CAP severity and help in determining whether a SCAP patient needs hospitalization or ICU admission. These include the CURB (confusion, blood urea > 42.8 mg/dL, respiratory rate > 30/min, blood pressure < 90/60 mmHg) and CURB-65 (confusion, blood urea > 42.8 mg/dL, respiratory rate > 30/min, blood pressure < 90/60 mmHg, age > 65 years), and the Pneumonia Severity Index (PSI). All these guidelines and measures have attempted to address CAP severity [4,5,6,7]. These two scoring rules have also been used to predict in-hospital mortality in patients with CAP [8]; however, the predictive accuracy is still in doubt [9].

In the ICU setting, models such as the Acute Physiology and Chronic Health Evaluation (APACHE) II score, Sequential Organ Failure Assessment (SOFA) score, and quick Sepsis Related Organ Failure Assessment (qSOFA) score have been designed to predict mortality in critically ill patients [10,11,12,13]. A perfect scoring model to predict outcomes requires precise data on the severity of illness and the associated risk of death. The APACHE II, SOFA, and qSOFA scores are good predictors of ICU mortality in patients with sepsis, and the APACHE II score is also an excellent independent predictor of mortality outcome in ventilator-associated pneumonia patients [14]. The APACHE II score was also applied to predict 30-day mortality in patients with SCAP; however, the predictive accuracy was also in doubt [15].

Malnutrition status may increase the risk of mortality in critically ill patients. Therefore, scores for disease severity have been developed to estimate the nutritional status in critically ill patients. The Nutrition Risk in Critically Ill (NUTRIC) score is an important nutritional risk assessment tool for ICU patients and its components of variables including the APACHE II score, SOFA score, number of comorbidities, days from hospital admission to ICU admission, and serum interleukin-6 (IL-6) level [16]. The NUTRIC score is a perfect tool for predicting clinical outcomes in critically ill patients; however, IL-6 should not be measured routinely in the ICU setting. Therefore, a modified NUTRIC (mNUTRIC) score was developed that included all variables except for the IL-6 level [17]. Although mNUTRIC score does not include any nutrition-related factor, however, many studies reported that early and urgent nutritional support may reduce mortality in high mNUTRIC score patients [18], reflecting the mNUTRIC score can represent the clinical nutrition status. In addition, the mNUTRIC score has been widely employed for screening clinical outcomes in critically ill patients, and its predictive accuracy for the 28-day mortality in septic patients is not inferior to the NUTRIC score [19]. The mNUTRIC score has also been proven as an effective tool for predicting ICU mortality in critically ill patients, including those with severe pneumonia [20] and sepsis [21]; however, studies have always declared small sample sizes and need further large sample size studies to elaborate the powerful evidence [19,22,23,24].

We conducted a large sample size study in a multi-center setting and aimed to determine if the mNUTRIC score would have the best prognostic accuracy than other important clinical prediction rules mentioned above for predicting hospital mortality and clinical treatment outcomes in intubated patients with SCAP.

## 2. Material and Methods

### 2.1. Setting and Study Design

This study is part of a multi-center study of the efficacy and safety of Brosym and Tazocin in the treatment of SCAP and nosocomial pneumonia. It is a retrospective, multi-center study (BATTLE study) [25,26], with a specific focus on the mNUTRIC score and clinical prediction rules for predicting clinical outcomes in patients with SCAP. We enrolled mechanically ventilated SCAP patients who were treated in a pulmonary ICU department in a multi-center setting (nine centers) between March 2018 and May 2019. All patients were admitted to the ICU ward for CAP diagnosis. Survival status and clinical treatment outcomes were recorded after the patient was discharged from the hospital. Patients were divided into two groups according to definitions: survivors and non-survivors, clinical cure, and clinical failure.

### 2.2. Definition

Patient files were reviewed to assess whether the study inclusion and exclusion criteria were fulfilled (Table 1) to judge the selection bias in a retrospective study. In brief, the diagnosis of pneumonia was made according to the newly occurred or progressive radiographic pulmonary infiltration/consolidation in patients with ≥two features of the following: cough, hyperthermia, hypothermia, purulent sputum, or respiratory secretion, and pathological lung auscultation. Patients registered at the first manifestation with CAP could be included in the study. 

We defined the clinical cure as the absence of signs or symptoms of pneumonia after completion of the antibiotic treatment course. Treatment failure indicated persistent infection or unimproved clinical condition and needed switching to another antibiotic regimen after 3 days of starting treatment. Hospital survivors were defined as patients who were still alive after discharge from admission due to pneumonia. Hospital non-survivors were referred to mortality due to any reason during a hospital stay. Successful treatment outcome is not certainly equal to hospital survival and unsuccessful treatment outcome is also not contributed to hospital non-survivor absolutely. 

### 2.3. Data Record and Evaluation

Data that was needed for the calculation of severity scores were evaluated on the first day of ICU admission. Pneumonia scoring systems (i.e., PSI and CURB-65) were calculated from the evaluated databases of the patients. The ICU scoring systems (i.e., qSOFA, SOFA, and APACHE II) were assessed during the ICU stay. To calculate the Charlson comorbidity index (CCI), information regarding the underlying comorbidities was collected from medical records and patient history. The mNUTRIC score data were calculated from the evaluated variables and patient databases. For identification of the possible responsible pathogen, microbiological analysis was performed using an endotracheal aspirate sample. Cultures from the blood samples were obtained before antibiotic administration. Bacterial pathogens isolated from sputum and blood were listed as risk factors for clinical outcome analysis. Antibiotic treatment duration and length of hospital stay may also influence clinical outcomes in patients with SCAP, and we recorded these duration times and included them in the clinical outcome risk factor analysis with other clinical prediction rules.

### 2.4. Ethics

All data were collected retrospectively after getting approval from ethics committees or institutional review boards of each hospital. Because the data were gathered on a routine basis, and the evaluation was conducted retrospectively, informed consent was not necessary by the approving ethics committees. All methods, including patient data confidentiality, were performed in accordance with the Declaration of Helsinki.

### 2.5. Statistical Analysis

We analyzed categorical variables by using the chi-square test, and continuous variables by using the Student’s *t*-test. Multivariate logistic regression analysis was performed to identify the most independent factors for hospital mortality and treatment outcomes. Factors that were put into the multivariate model analysis included the mNUTRIC score, all clinical prediction rules, and other clinical factors that showed statistical significance by univariate analysis. Receiver operating characteristic (ROC) curves were plotted, and the area under the curve (AUC) was compared with the clinical prediction rules measured in this study. The cut-off values of all factors for predicting hospital mortality and treatment outcomes among patients with SCAP were analyzed according to the ROC curves.

We also wanted to see the relevance between clinical factors and clinical prediction rules that most influence clinical outcomes. Thus, after analysis of the statistically significant factors for predicting clinical outcomes, we analyzed the correlation between the clinical factors and clinical prediction rules using Student’s *t*-test and chi-square methods.

The results are expressed as absolute numbers (percentages) or mean ± standard deviation. Adjusted odds ratios and 95% confidence intervals (CIs) were calculated using the multivariate logistic regression method. Statistical significance was set at *p* < 0.05. All statistical analyses were measured using the SPSS software (version 22; SPSS Inc., Chicago, IL, USA).

## 3. Results

### 3.1. Population Characteristics

A total of 1225 intubated patients with severe pneumonia were found during the study period; 410 patients were diagnosed as nosocomial pneumonia and excluded from the study, while 815 patients were diagnosed as SCAP and enrolled in this study. They included 581 male and 134 female patients, with a mean age of 76.82 ± 14.57 years. In this study group, 137 patients died, and 678 patients were discharged. The in-hospital mortality rate was 16.8%. Regarding treatment outcomes, 667 patients were clinically cured, and 148 patients experienced treatment failure (Figure 1). A total of 105 patients with bacteremia were identified in our study cohort, and the prevalence of bacteremia was 12.88%.

### 3.2. Comparison of Predictive Accuracy for Hospital Mortality

In this study, CCI (*p* < 0.001), PSI score (*p* < 0.001), CURB-65 score (*p* < 0.001), APACHE II score (*p*< 0.001), SOFA score (*p* = 0.031), and mNUTRIC score (*p* < 0.001) were significantly higher in non-survivors than in survivors. There was no significant difference between the qSOFA scores of non-survivors and survivors. We also found that longer antibiotic treatment duration *(p* = 0.011) and bacteremia status (*p* = 0.001) were significantly different between non-survivors and survivors. Pathogen analysis revealed that *S. aureus* isolated from blood (*p* = 0.001) and sputum (*p* = 0.001) were also more frequently found in non-survivors than in survivors (Table 2). Using the ROC method, curves were plotted to identify cut-off values that would best determine hospital mortality. We found that the mNUTRIC score (*p* < 0.001) and APACHE II score (*p* = 0.001) were highly accurate in predicting hospital mortality and reached statistical significance. The optimal cut-off value for the APACHE II score and the AUC value was 23.5 and 0.785, respectively, and yielded a sensitivity and specificity of 66.7% and 77.4%, respectively. Beyond the APACHE score, the mNUTRIC score had a higher AUC value (0.838), and the optimal cut-off value was 5.5, yielding a sensitivity and specificity of 80.0% and 77.4%, respectively (Figure 2, Table 3).

### 3.3. Comparison of Predictive Accuracy for Treatment Outcome

Regarding treatment outcomes for SCAP patients, we also found that CCI (*p* < 0.018) and PSI (*p* < 0.001), CURB-65 (*p* < 0.001), APACHE II (*p* = 0.001), SOFA (*p* = 0.040), and mNUTRIC scores (*p* < 0.001) were significantly higher in patients with clinical cure than in those with clinical failure. Pathogen analysis revealed that *S. aureus* isolated from blood (*p* = 0.010) was more likely to result in clinical treatment failure than no isolates from blood (Table 4). For the ROC curve analysis of the predictability of treatment outcomes between different clinical prediction rules and mNUTRIC score, we found that mNUTRIC (*p* = 0.001) and APACHE II score (*p* = 0.008) had high accuracy for predicting treatment outcomes, with significance. The optimal cut-off and AUC values for the APACHE II score were 23.5% and 0.727, respectively, showing a sensitivity and specificity of 57.1% and 74.6%, respectively. Compared to the APACHE II score, the mNUTRIC score revealed a higher AUC value (0.773), and the optimal cut-off value was 5.5, with a sensitivity and specificity of 78.6% and 76.2%, respectively (Figure 3, Table 5).

### 3.4. Predictors for Hospital Mortality and Treatment Outcome

In this study, we used multivariate logistic regression to determine independent clinical factors for predicting hospital mortality and treatment outcomes. We found that bacteremia status (*p* = 0.030, OR = 0.065 (0.006–0.772)) and mNUTRIC score (*p* = 0.003, OR = 2.954 (1.457–5.991)) were independent predictors of hospital mortality. We also found that the APACHE II score (*p* = 0.036, OR = 1.197 (1.012–1.416)), mNUTRIC score (*p* = 0.019, OR = 1.848 (1.107–3.086)), and duration of antibiotic treatment (*p* = 0.047, OR = 1.107 (1.001–1.224)) were independent factors for predicting treatment outcomes for clinical cure (Table 6).

### 3.5. Relevance between Clinical Factors and Clinical Prediction Rules

After analysis, we found that bacteremia status and mNUTRIC score were independent factors for hospital mortality. Duration of antibiotic treatment, APACHE II score, and mNUTRIC score were independent factors for clinical treatment outcomes. We analyzed the correlations between these factors. We found that bacteremia status (*p* < 0.001) and duration of antibiotic treatment (*p* < 0.001) were highly correlated with the mNUTRIC and APACHE II scores. The length of hospital stay showed no correlation with the mNUTRIC and APACHE II scores (Table 7).

## 4. Discussion

In this study, we enrolled 815 patients with SCAP and reported four main findings. First, we found that applying scoring in clinical prediction rules, such as CCI or PSI, CURB-65, APACHE II, SOFA, and mNUTRIC scores, could determine hospital mortality and treatment outcomes. Second, prolonged antibiotic administration duration had no effect on hospital mortality or treatment outcomes. Third, only the mNUTRIC score was an independent determinant of hospital mortality and treatment outcomes. The mNUTRIC score cut-off value was 5.5 for these two outcomes. Fourth, we also found that *S. aureus* isolated from blood or sputum may result in hospital mortality and clinical treatment failure.

Critical illness with malnutrition would contribute to an aggravated progression of nosocomial infection, expanded hospital stays, difficulty in weaning, and finally resulting in substantial morbidity and mortality. As literature reported, malnutrition status could lead to the imbalance of pro-inflammatory and anti-inflammatory adipokines and contribute to insulin resistance that makes diabetes and metabolic disease worse [27]. Furthermore, the immunologic system would be dampened and provoke a serious critical situation. Intensive sugar control helping the all-cause mortality in critically ill patients is well-known [28] and proper nutrition support may also restore the immunologic and biochemical balance and create a favorable clinical outcome. Since diabetes and metabolic syndrome may initiate chronic inflammation and induce excess oxidative stress, authors have reported diet therapy and nutrition treatment for obstructive sleep apnea (OSA) and diabetic neuropathy. Ghitea et al. reported that diet therapy with exercise would improve metabolic syndrome and ameliorate OSA patients’ outcomes [29]. Bondar et al. reported that pathogenic treatment with α-lipoic acid and benfotiamine are important therapies for diabetic neuropathy [30]; they all emphasized proper nutrition management would weaken oxidative stress and inflammation, further perfect the immunologic system, and improve patient’s prognosis. The inseparable association between malnutrition and mortality is really presented. Malnutrition mainly contributes to mortality by decomposing the immunologic system and increasing the susceptibility to infectious diseases. The severity of critical disease would aggravate if once malnourished individuals infected. In addition, if important infectious diseases are present, malnutrition may lead to more mortality compared to a circumstance in which individuals are not commonly infected by these organisms. Early nutritional support in critically ill patients at high risk of malnutrition may improve patient outcomes [31,32]. Many studies have reported that the mNUTRIC score is a perfect tool for malnutrition status screening in critically ill patients, which may lead to undesirable outcomes [19,20,33,34] despite the mNUTRIC score does not include any nutrition-related factor. The mNUTRIC score is based on APACHE II, SOFA score, and comorbidities and serves as a more effective tool for predicting clinical outcomes. In this study, we also verified that the mNUTRIC score has better predictive accuracy for predicting hospital mortality and treatment outcome in intubated patients with SCAP compared to other scoring systems. The mNUTRIC score of >5 indicated a high nutritional risk status and urgently requiring additional energy support to reduce the likelihood of mortality [18]. Our study also validated this rule; moreover, we reported a cut-off value of 5.5.

Some studies have evaluated the availability of these scoring systems and have reported different findings. The PSI score is a clinical prediction rule that medical clinicians can easily use to determine the possibility of ICU admission in patients with SCAP. The PSI score is often used to estimate the necessary hospitalization in pneumonia patients and is more precise in verifying low-risk pneumonia patients who are potential candidates for outpatient care [35,36,37]. Few studies have validated the PSI score for predicting ICU or hospital mortality. The CURB-65 score is also a well-known clinical prediction rule that has been validated for predicting mortality in CAP [6]. Many studies have addressed the CURB-65 score, focusing mainly on the outcomes of ICU admission, hospital admission, or outpatient care. However, CURB-65 scores have not previously been assessed for predicting mortality in patients with CAP on mechanical ventilators [38,39,40,41,42,43,44,45,46]. In our study, we evaluated these scoring systems and found that the PSI and CURB-65 scores can also correlate with hospital mortality and treatment outcomes in intubated patients with SCAP. However, these two rules were correlated with clinical outcomes, and their predictive accuracy was lower than the mNUTRIC score by multivariate and ROC methods.

The APACHE II, SOFA, and qSOFA scores were designed to predict mortality in critically ill patients. In our study, we also found that APACHE II and SOFA scores were correlated with hospital mortality and treatment outcomes. Multivariate analysis revealed that an APACHE II score of >23.5 was an independent predictor of mortality. Using the ROC method, we also observed that the APACHE II score had a high predictive accuracy for hospital mortality and treatment outcome with good AUC value, good sensitivity, and specificity. The APACHE II score is a famous and validated prognostic indicator for management in ICU. In addition, the APACHE II severity score has indicated good calibration and discriminatory value for disease processes and is one of the most used severity scoring systems worldwide [47]. Based on APACHE II and SOFA scores, the mNUTRIC score illustrated a better predictive accuracy and was an independent factor for hospital mortality and treatment outcome in intubated patients with SCAP by multivariate and ROC methods.

The CCI has been commonly used to evaluate the importance of comorbidities in a variety of critical situations. This index was created in 1987 and is a prognostic scoring method that has been validated in many clinical settings [48,49,50]. For example, the CCI was able to predict prognosis in patients with chronic lung disease [51], in non-surgical patients visiting the emergency room [52], and in elderly CAP patients admitted to an acute care hospital [53]. Accordingly, in our study, CCI was also correlated with hospital mortality and treatment outcomes in patients with SCAP and mechanical ventilation. Furthermore, the number of comorbidities is always a variable of the mNUTRIC score, and the mNUTRIC score demonstrated better predictive accuracy than the CCI score.

The mNUTRIC score is an effective tool for detecting malnutrition among ICU patients, as previously reported [54]. Malnutrition status in the ICU is linked with inflammation and a hypermetabolic state and may further result in decreased immunity, muscle wasting, poor wound healing, and deteriorating infectious processes [55]. In our study, bacteremia status was also an independent factor for hospital mortality, and bacteremia with S. aureus isolates would contribute to hospital mortality and clinical treatment failure. Community-acquired S. aureus pneumonia appears to be a severe disease that easily results in multilobar shadowing, high ICU admission rate, and high hospital mortality as reported [56] in a previous article, with most infections containing Panton Valentine leukocidin (PVL) genes and were uniformly resistant to general empiric antibiotic treatment [57]. We also found that bacteremia status was positively correlated with mNUTRIC and APACHE II scores. Bacteremia status refers to critical infectious severity and always requires prolonged antibiotic treatment. In this study, prolonged antibiotic administration duration correlated with hospital mortality. This does not mean that prolonged antibiotic treatment is harmful and simply indicates that the disease severity is profound. Our study also showed that prolonged antibiotic treatment was correlated with the mNUTRIC and APACHE II scores. Furthermore, prolonged antibiotic administration duration had no effect on hospital mortality and treatment outcomes, as reported in a previous study [58], and most emphasis was based on malnutrition status and disease severity.

The strengths of our study were that its multi-center design and inclusion of large amount patients; however, there were still some limitations to this study. First, this was a retrospective and observational study in which data were retrieved from electronic medical records, and the non-randomized nature limited the interpretation of the results of the study. Second, there were some missing data from some patients because of the retrospective design, especially laboratory values. Third, owing to the multi-center design, the differences in reporting manner and clinical management could have existed; however, there was also increased generalizability of this study’s findings. For example, clinicians could select antimicrobial regimens in a self-determination manner, which may introduce a degree of bias and render the direct comparison of clinical outcomes difficult. Clinicians usually managed critically ill patients with carbapenem, which could influence the mortality rate of SCAP patients.

## 5. Conclusions

We found that the mNUTRIC score was a better independent factor for predicting hospital mortality and treatment outcomes than other scoring systems. In this study, the mNUTRIC score cut-off value was 5.5.

## Figures and Tables

**Figure 1 nutrients-14-00198-f001:**
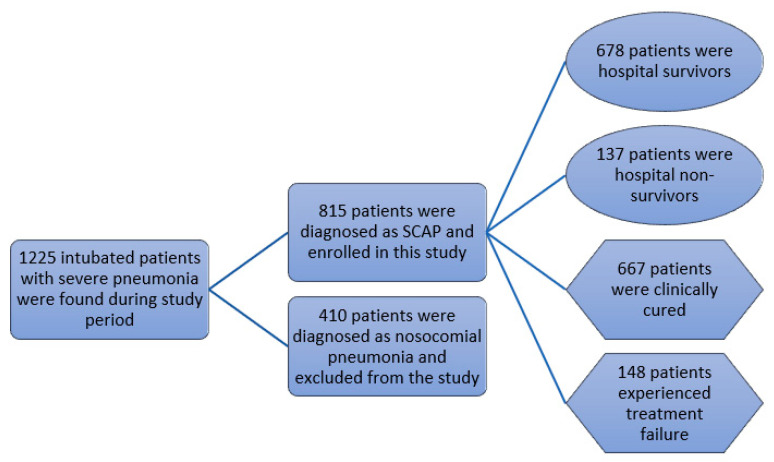
Flow diagram of population characteristics.

**Figure 2 nutrients-14-00198-f002:**
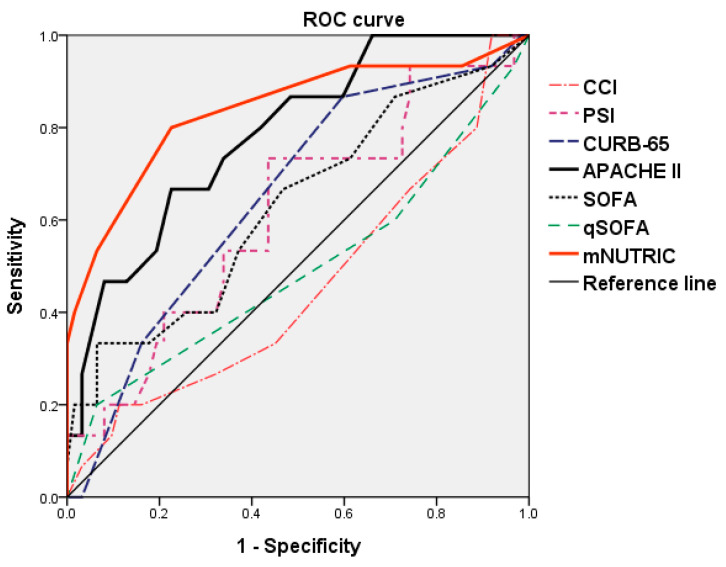
ROC curve analysis for predictability of mortality between different clinical prediction rules.

**Figure 3 nutrients-14-00198-f003:**
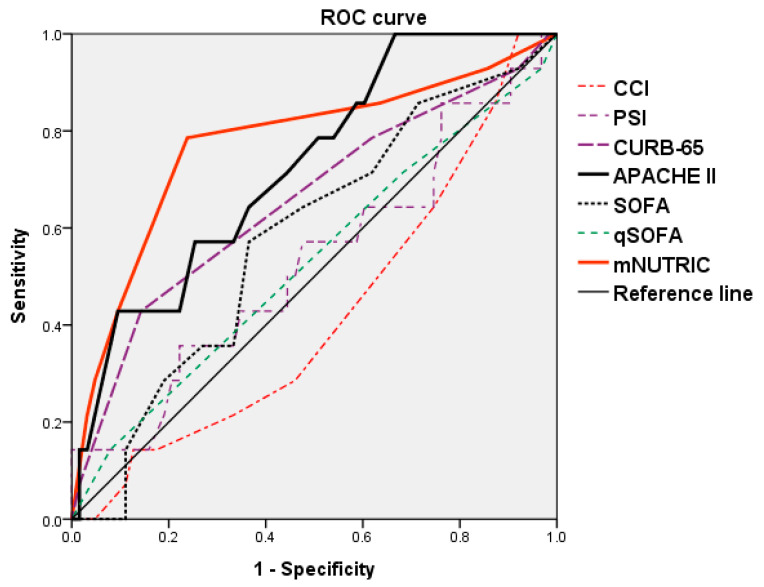
ROC curve analysis for predictability of treatment outcome between different clinical prediction rules.

**Table 1 nutrients-14-00198-t001:** Inclusion and exclusion criteria of community-acquired pneumonia.

Inclusion Criteria	Exclusion Criteria
• Age ≥18 years • Newly developed or progressive radiographic lung infiltration/consolidation (confirmed by radiologist) • At least two of the following symptoms of lower respiratory tract infection Cough Purulent expectoration Fever ≥38.3 °C or hypothermia <35 °C Pathological lung auscultation	• Pneumonia developed 48 h after admission or intubation; • Pneumonia developed at a nursing home; • Hospitalization in the last 30 days; • Immunosuppressed state (i.e., treatment with steroids, cytotoxic agents, and/or immunosuppressive agents for longer than 1 month).

**Table 2 nutrients-14-00198-t002:** Characteristics of clinical prediction rules and clinical factors for predicting hospital mortality in patients with SCAP.

	Survivors (*n* = 678)	Non-Survivors (*n* = 137)	*p*-Value
Age	76.72 ± 14.25	76.48 ± 16.14	0.860
Sex Male Female	496 (73.16%) 182 (26.85%)	85 (62.04%) 52 (37.96%)	0.007 **
mNUTRIC score	4.78 ± 1.20	6.87 ± 1.72	<0.001 ***
CCI	5.76 ± 2.54	6.72 ± 3.25	<0.001 ***
PSI score	142.92 ± 42.61	165.56 ± 46.11	<0.001 ***
CURB-65 score	2.55 ± 1.25	2.99 ± 1.31	<0.001 ***
APACHE II score	20.70 ± 6.86	25.89 ± 7.10	<0.001 ***
SOFA	6.36 ± 3.45	7.86 ± 4.10	0.031 *
qSOFA	1.79 ± 0.64	1.75 ± 0.86	0.844
Treatment duration	9.03 ± 3.36	10.12 ± 4.78	0.011 *
Length of hospital stay	29.00 ± 13.09	28.86 ± 20.24	0.977
Bacteremia Yes No	76 (11.21%) 602 (88.79%)	29 (21.17%) 108 (78.83%)	0.001 **
Pathogens of blood culture Streptococcus pneumonia Haemophilus influenzae Moraxella catarrhalis Serratia marcescens Escherichia coli Klebsiella pneumonia Staphylococcus aureus	14 (2.1%) 5 (0.7%) 6 (0.9%) 10 (1.5%) 16 (2.4%) 18 (2.7%) 7 (1.0%)	5 (3.6%) 1 (0.7%) 1 (0.7%) 3 (2.2%) 5 (3.6%) 6 (4.4%) 8 (5.8%)	0.345 0.734 0.666 0.467 0.376 0.270 0.001 **
Sputum culture Growth No Growth	322 (47.49%) 356 (52.51%)	65 (47.45%) 72 (52.55%)	0.508
Pathogens of sputum culture Streptococcus pneumonia Haemophilus influenza Moraxella catarrhalis Serratia marcescens Escherichia coli Klebsiella pneumonia Staphylococcus aureus	91(13.4%) 40 (5.9%) 24 (3.5%) 36 (5.3%) 31 (4.6%) 62 (9.2%) 38 (5.6%)	19 (13.9%) 4 (2.9%) 4 (2.9%) 8 (5.8%) 5 (3.6%) 6 (4.4%) 19 (13.9%)	0.490 0.110 0.479 0.466 0.418 0.088 † 0.001 **

*** *p* < 0.001; ** *p* < 0.01; * *p* < 0.05; † *p* < 0.09; Abbreviations: APACHE: Acute Physiology and Chronic Health; Evaluation; CCI: Charlson comorbidity index; CURB-65: Confusion, blood urea > 42.8 mg/dL, respiratory rate > 30/min, blood pressure < 90/60 mmHg, age > 65); mNUTRIC: modified Nutrition Risk in the Critically Ill; PSI: Pneumonia severity index; qSOFA: quick Sepsis Related Organ Failure Assessment; SOFA: Sequential Organ Failure Assessment.

**Table 3 nutrients-14-00198-t003:** Comparison of the cut-off value, sensitivity, specificity, AUC, and *p*-value between severity scoring factors for hospital mortality.

Scoring Factors	Sensitivity	Specificity	Cut-Off Value	AUC	*p*-Value
CCI	0.200	0.887	9.5	0.458	0.611
PSI	0.733	0.565	143	0.615	0.171
CURB-65	0.867	0.403	2.5	0.652	0.070 †
APACHE II	0.667	0.774	23.5	0.785	0.001 **
SOFA	0.667	0.532	4.5	0.631	0.117
qSOFA	0.200	0.935	2.5	0.494	0.938
mNUTRIC	0.800	0.774	5.5	0.838	<0.001 ***

*** *p* < 0.001; ** *p* < 0.01; † *p* < 0.09; *p*-value is generated from the output of SPSS software by ROC method. Abbreviations: APACHE: Acute Physiology and Chronic Health; Evaluation; AUC: Area under the curve; CCI: Charlson comorbidity index; CURB-65: Confusion, blood urea > 42.8 mg/dL, respiratory rate > 30/min, blood pressure < 90/60 mmHg, age > 65); mNUTRIC: modified Nutrition Risk in the Critically Ill; PSI: Pneumonia severity index; qSOFA: quick Sepsis Related Organ Failure Assessment; ROC: Receiver operating characteristic; SOFA: Sequential Organ Failure Assessment.

**Table 4 nutrients-14-00198-t004:** Characteristics of clinical prediction rules and clinical factors for predicting treatment outcome in patients with SCAP.

	Clinical Cure (*n* = 667)	Failure (*n* = 148)	*p*-Value
Age	76.86 ± 14.45	76.54 ± 15.60	0.824
Sex Male Female	490 (73.46%) 177 (26.54%)	91 (61.49%) 57 (38.51%)	0.006 **
mNUTRIC score	4.93 ± 1.38	6.03 ± 1.78	<0.001 ***
CCI	5.79 ± 2.59	6.40 ± 2.95	0.018 *
PSI score	141.46 ± 41.64	176.64 ± 45.25	<0.001 ***
CURB-65 score	2.50 ± 1.22	3.41 ± 1.24	<0.001 ***
APACHE II score	20.72 ± 6.98	25.44 ± 6.80	0.001 **
SOFA	6.25 ± 3.58	7.73 ± 3.60	0.040 *
qSOFA	1.78 ± 0. 67	1.77 ± 0.83	0.943
Treatment duration	9.03 ± 3.33	9.81 ± 4.64	0.074 †
Length of hospital stay	26.19 ± 11.25	32.00 ± 19.48	0.161
Bacteremia Yes No	77 (11.54%) 590 (88.46%)	28 (18.92%) 120 (81.08%)	0.069 †
Pathogens of blood culture Streptococcus pneumonia Haemophilus influenza Moraxella catarrhalis Serratia marcescens Escherichia coli Klebsiella pneumonia Staphylococcus aureus	15 (2.2%) 5 (0.7%) 6 (0.9%) 9 (1.3%) 17 (2.5%) 18 (2.7%) 7 (1.0%)	4 (2.7%) 1 (0.7%) 1 (0.7%) 4 (2.7%) 4 (2.7%) 6 (4.1%) 8 (5.4%)	0.763 0.701 0.628 0.269 0.548 0.417 0.010 *
Sputum culture Growth No Growth	312 (46.78%) 355 (53.22%)	75 (50.68%) 73 (49.32%)	0.441
Pathogens of sputum culture Streptococcus pneumonia Haemophilus influenza Moraxella catarrhalis Serratia marcescens Escherichia coli Klebsiella pneumonia Staphylococcus aureus	90 (13.5%) 35 (5.2%) 21 (3.1%) 34 (5.1%) 30 (4.5%) 59 (8.8%) 43 (6.4%)	20 (13.5%) 9 (6.1%) 7 (4.7%) 10 (6.8%) 6 (4.1%) 9 (6.1%) 14 (9.5%)	0.542 0.688 0.323 0.422 0.511 0.533 0.212

*** *p* < 0.001; ** *p* < 0.01; * *p* < 0.05; † *p* < 0.09; Abbreviations: APACHE: Acute Physiology and Chronic Health; Evaluation; CCI: Charlson comorbidity index; CURB-65: Confusion, blood urea > 42.8 mg/dL, respiratory rate > 30/min, blood pressure < 90/60 mmHg, age > 65); mNUTRIC: modified Nutrition Risk in the Critically Ill; PSI: Pneumonia severity index; qSOFA: quick Sepsis Related Organ Failure Assessment; SOFA: Sequential Organ Failure Assessment.

**Table 5 nutrients-14-00198-t005:** Comparison of cut-off value, sensitivity, specificity, AUC, and *p*-value between severity scoring factors for treatment outcome.

Scoring Factors	Sensitivity	Specificity	Cut-Off Value	AUC	*p*-Value
CCI	0.143	0.873	9.5	0.422	0.366
PSI	0.143	1.000	219	0.530	0.726
CURB-65	0.429	0.857	3.5	0.657	0.067 †
APACHE II	0.571	0.746	23.5	0.727	0.008 **
SOFA	0.571	0.635	5.5	0.580	0.352
qSOFA	0.143	0.921	2.5	0.529	0.731
mNUTRIC	0.786	0.762	5.5	0.773	0.001 **

** *p* < 0.01; † *p* < 0.09; *p*-value is generated from the output of SPSS software by ROC method. Abbreviations: APACHE: Acute Physiology and Chronic Health; Evaluation; AUC: Area under the curve; CCI: Charlson comorbidity index; CURB-65: Confusion, blood urea > 42.8 mg/dL, respiratory rate > 30/min, blood pressure < 90/60 mmHg, age > 65); mNUTRIC: modified Nutrition Risk in the Critically Ill; PSI: Pneumonia severity index; qSOFA: quick Sepsis Related Organ Failure Assessment; ROC: Receiver operating characteristic; SOFA: Sequential Organ Failure Assessment.

**Table 6 nutrients-14-00198-t006:** Determination of predictors for clinical outcomes among 815 SCAP patients by multivariable logistic regression analysis.

	Hospital Mortality AOR (95% CI)	*p*-Value	Treatment Outcome AOR (95% CI)	*p*-Value
CCI	1.194 (0.813–1.753)	0.365	0.924 (0.623–1.371)	0.695
PSI score	1.018 (0.982–1.056)	0.324	0.985 (0.955–1.016)	0.344
CURB-65 score	1.410 (0.371–5.353)	0.614	2.257 (0.702–7.259)	0.172
APACHE II score	1.240 (0.995–1.544)	0.056 †	1.197 (1.012–1.416)	0.036 *
SOFA score	0.856 (0.611–1.200)	0.368	0. 762 (0.555–1.047)	0.093
qSOFA score	0.800 (0.158–4.054)	0.788	1.569 (0.438–5.622)	0.489
mNUTRIC score	2.954 (1.457–5.991)	0.003 **	1.848 (1.107–3.086)	0.019 *
Bacteremia	0.065 (0.006–0.772)	0.030 *	0.604 (0.065–5.655)	0.659
Treatment duration	1.246 (0.977–1.588)	0.076 †	1.100 (0.996–1.215)	0.047 *

** *p* < 0.01; * *p* < 0.05; † *p* < 0.09; Abbreviations: AORs: Adjusted odds ratios; APACHE: Acute Physiology and Chronic Health Evaluation; CCI: Charlson comorbidity index; CIs: Confidence intervals; CURB-65: Confusion, blood urea > 42.8 mg/dL, respiratory rate > 30/min, blood pressure < 90/60 mmHg, age > 65); mNUTRIC: modified Nutrition Risk in the Critically Ill; PSI: Pneumonia severity index; qSOFA: quick Sepsis Related Organ Failure Assessment; SOFA: Sequential Organ Failure Assessment.

**Table 7 nutrients-14-00198-t007:** Relevance between bacteremia, treatment duration, length of hospital stay, mNUTRIC score, and APACHE II score.

	mNUTRIC Score		*p*-Value	APACHE II Score		*p*-Value
	>5 *N* = 333	≤5 *N* = 481		>23.5 *N* = 347	≤23.5 *N* = 467	
Bacteremia Yes No	70 (21.02%) 263 (78.98%)	35 (7.27%) 446 (92.72%)	<0.001 ***	72 (20.75%) 275 (79.25%)	33 (7.07%) 434 (92.93%)	<0.001 ***
Treatment duration	10.53 ± 4.53	8.30 ± 2.53	<0.001 ***	9.88 ± 4.37	8.72 ± 2.94	<0.001 ***
Length of hospital stay	28.71 ± 15.68	29.71 ± 16.88	0.838	29.24 ± 16.00	27.50 ± 15.71	0.754

*** *p* < 0.001; Abbreviations: APACHE: Acute Physiology and Chronic Health Evaluation; mNUTRIC: modified Nutrition Risk in the Critically Ill.

## Data Availability

The data presented in this study are available on request from the corresponding author. The data are not publicly available due to patients’ privacy.

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
