# Peer review of "Significance of the Modified NUTRIC Score for Predicting Clinical Outcomes in Patients with Severe Community-Acquired Pneumonia"

_nutrients, 2021, doi:10.3390/nu14010198_

Round 1

Reviewer 1 Report

The article presents a not commonly used score for predicting mortality in pneumonia. I appreciate the relevance of your results and he graphical presentation which makes it very clear that this score is usefull for the practitioner, the methods are accurate, the statisticall analysis is correct. Congratulations!

Please improve a bit the discussions by commenting how does diabetes mellitus and metabolic diseases impact the risk of pro-inflammatory syndrome with rising of certain cytokines that cand impact the value of your score.

Please check:

  1. Ghitea TC, Aleya L,  Pallag A, Bungau S. Influence of diet and sport on the risk of sleep apnea in patients with metabolic syndrome associated with hypothyroidism - a 4-year survey. Environ Sci Pollut Res Int. 2021 Nov 20.  doi: 10.1007/s11356-021-17589-x.
  2. Bondar, A., Popa, A. R., Papanas, N., Popoviciu, M., Vesa, C. M., Sabau, M., Daina, C., Stoica, R. A., Katsiki, N., Stoian, A. P. Diabetic neuropathy: A narrative review of risk factors, classification, screening and current pathogenic treatment options (Review). Experimental and Therapeutic Medicine 22, no. 1 (2021): 690.  https://doi.org/10.3892/etm.2021.10122 

Author Response

Response to Reviewer 1 Comments:

Point: Please improve a bit the discussions by commenting how does diabetes mellitus and metabolic diseases impact the risk of pro-inflammatory syndrome with rising of certain cytokines that can impact the value of your score.

Please check:

Ghitea TC, Aleya L,  Pallag A, Bungau S. Influence of diet and sport on the risk of sleep apnea in patients with metabolic syndrome associated with hypothyroidism - a 4-year survey. Environ Sci Pollut Res Int. 2021 Nov 20.  doi: 10.1007/s11356-021-17589-x.

Bondar, A., Popa, A. R., Papanas, N., Popoviciu, M., Vesa, C. M., Sabau, M., Daina, C., Stoica, R. A., Katsiki, N., Stoian, A. P. Diabetic neuropathy: A narrative review of risk factors, classification, screening and current pathogenic treatment options (Review). Experimental and Therapeutic Medicine 22, no. 1 (2021): 690.  https://doi.org/10.3892/etm.2021.10122

Response: We thank reviewer’s encouragement and valuable comment. We have state the correlation between diabetes mellitus, metabolic syndrome and malnutrition in the Discussion section (page 10) and describe that proper nutrition support could diminish the oxidative stress and inflammation status in malnourished patients. We also cite the suggestive references. We believe this change would improve the value our manuscript. Thanks a lot!

Reviewer 2 Report

The manuscript describes a well-designed study that contributes important findings to the field. I offer several suggestions for moderate revision:

Line 160- Population characteristics. A flow diagram would be valuable to depict the process of excluding participants from initial to final number.

Lines 161-162: insert "the" study.

Figure 1 & Figure 2 The colored lines are difficult to distinguish. Suggest adding symbols or dashed/dotted lines to make the identities more clear.

Table 3 & Table 5: Describe in the footnote the comparisons used to generate the p values. The titles are appropriate, but the analysis associated with the p value ought to be stated in the footnote.

Line 239: The treatment duration effect p value in the text (p=0.047) and table (p=0.60) are different.

Lines 279-281: Sentence structure needs correction.

Line 328: change to "effective tool for detecting malnutrition..."

In general, please describe the mNUTRIC as "better than" other scoring systems rather than "the best."

Author Response

Response to Reviewer 2 Comments:

Point1: Line 160- Population characteristics. A flow diagram would be valuable to depict the process of excluding participants from initial to final number.

Response 1: We have plotted a flow diagram of population characteristics (figure 1, page 5). 

Point 2: Lines 161-162: insert "the" study.

Response 2: We have inserted “the” study as suggestion (page 5).

Point 3: Figure 1 & Figure 2 The colored lines are difficult to distinguish. Suggest adding symbols or dashed/dotted lines to make the identities more clear.

Response 3: We have re-plotted these two figures (figure 2, figure 3) and we use color and dashed/dotted lines to distinguish each scoring items. We believe this change would make the identities more clear. Thanks a lot.

Point 4: Table 3 & Table 5: Describe in the footnote the comparisons used to generate the p values. The titles are appropriate, but the analysis associated with the p value ought to be stated in the footnote.

Response 4: We have added the comparisons of p- value in footnotes of each table and we also stated that p-value is generated from ROC method in table 3 and table 5. Thanks for reviewer’s suggestion. 

Point 5: Line 239: The treatment duration effect p value in the text (p=0.047) and table (p=0.60) are different.

Response 5: We thank reviewer’s attentiveness. We have corrected the error in table 6.

Point 6: Lines 279-281: Sentence structure needs correction.

Response 6: We have corrected the sentence as “In this study, we also verified that the mNUTRIC score has better predictive accuracy for predicting hospital mortality and treatment outcome in intubated patients with SCAP compared to other scoring systems.” (page 10)

Point 7: Line 328: change to "effective tool for detecting malnutrition..."

Response: We have amended the sentence (page 11). Thanks a lot

Point 8: In general, please describe the mNUTRIC as "better than" other scoring systems rather than "the best."

Response: We thank reviewer’s valuable comment. We have corrected the description of mNUTRIC score as “better than other scoring systems” rather than “the best” (page 1, page 10, page 12). 
